# Detection of Optic Disc Drusen in Children Using Ultrasound through the Lens and Avoiding the Lens—Point of Care Ultrasound Technique of Evaluation Revisited

**DOI:** 10.3390/jcm8091449

**Published:** 2019-09-12

**Authors:** Renuka Rajagopal, Ellen Mitchell, Christin Sylvester, Lea Ann Lope, Ken Kanwal Nischal

**Affiliations:** 1UPMC Eye Center, UPMC Children’s Hospital of Pittsburgh, Pittsburgh, PA 15224, USA; renukarajagopal2112@gmail.com (R.R.); mitchelleb@upmc.edu (E.M.); lopel@upmc.edu (C.S.); sylvestercl@upmc.edu (L.A.L.); 2University of Pittsburgh School of Medicine, Pittsburgh, PA 15224, USA

**Keywords:** optic nerve drusen, B scan, papilledema

## Abstract

Aim: To assess whether the detection rate of optic disc drusen (ODD) in children with swollen optic discs varies if the ultrasound scan (USS) is performed through the lens or avoiding the lens. Methods: Retrospective review of the ultrasound machine database for all patients who underwent USS for swollen discs in the department of pediatric ophthalmology, UPMC Children’s Hospital of Pittsburgh. Only patients who had both fundus pictures and USS performed (through and avoiding the lens) were included in the study. Results: A total of 31 patients (62 eyes) were included in the study. USS detected ODD in 44% of eyes (27 of 62 eyes, 15 patients). In 82% of these eyes (22 of 27 eyes), the ODD were not detected initially when scanning was done through the lens but were only detected when scanning was performed avoiding the lens. Ten out of sixteen patients with no ODD on USS had another identifiable cause for disc elevation, including raised intracranial pressure and sleep apnea. Conclusion: Ultrasound is a sensitive diagnostic tool for detecting ODD. The rate of detection of ODD is increased when USS is done avoiding the lens in children where the ODD are usually buried and not as calcified as those found in adults. Under such circumstances, the reduced echogenicity is absorbed by the absorbent pediatric lens, thus limiting the detection rates when scanning through the lens.

## 1. Introduction

It is vital to differentiate optic disc drusen (ODD) from papilledema to avoid unnecessary investigations in an otherwise asymptomatic patient [1]. Various modalities are available to diagnose ODD including ultrasound scan (USS), ocular coherence tomography (OCT), fluorescein angiography, and CT scan [2]. On OCT, ODD is seen as a hypoechogenic subretinal structure with hyperreflective margins [3]. The description of ODD on OCT is variable. While some studies describe drusen as a hypo reflective subretinal structure, some studies report drusen as a hyperreflective mass. Furthermore, these findings are not specific to ODD. For instance, there are cases of optic disc edema described to have hyperreflective margins similar to ODD. Spectral domain OCT comparing retinal nerve fiber layer (RNFL) thickness can be used to differentiate ODD from papilledema [4]. Kulkarni et al. [4] reported that the contour of elevation was smoother in optic disc edema compared to ODD. However, this test has low specificity and poor interobserver measurements [4]. Autofluorescence on fundus pictures can be useful in detecting ODD but is not as sensitive for buried drusen. USS is the gold standard in the diagnosis of ODD [2]. USS is a noninvasive tool and can be performed in pediatric patients easier than the other tests described above. On an ultrasound B-Mode scan, drusen are seen as hyperreflective structures on the optic nerve head visible with a gain of 0dB. However, on USS, the image resolution is affected due to attenuation by the lens [5]. This may not be significant in adults whose ODD are robustly calcified and found on the surface of their optic discs, but in children with buried ODD [6], reduced calcification and increased lens absorption are likely to prevent enough transmission of reflected waves from the optic nerve head to the transducer. Thus, it is desirable to scan avoiding the lens to improve the resolution of the images on USS [3]. While scanning protocols in adults using longitudinal, axial, and transverse positions of the USS probe should capture scans avoiding the lens, to the best of our knowledge, there are no studies in the literature specifically investigating the effect of negating lens attenuation by scanning while avoiding the lens to detect ODD. Furthermore, we describe a simple two-step scanning protocol that is easier to deploy in children than the protocol described above.

## 2. Materials and Methods

We retrospectively analyzed medical records of all patients who underwent ultrasonography for a swollen optic disc between 2012 and 2015 in the department of pediatric ophthalmology at the UPMC Children’s Hospital of Pittsburgh. The study gained IRB approval from the University of Pittsburgh research office and all work pertaining to this study was carried out in a HIPAA compliant manner.

Patients who had optic disc swelling on presentation and had an USS performed both through and avoiding the lens as well as fundus photography were included in the study. Ultrasound scans were performed in the horizontal axial, vertical axial, and vertical transverse planes using a 10 MHz linear ultrasound transducer for a conventional through the lens scan. Once the optic nerve head was imaged the gain was reduced to 0 dB and the presence or absence of drusen was noted. For the avoid-the-lens scan, patients were asked to keep their eyes open and look up. The probe was then placed on the lower lid with the marker on the horizontal plane and the eye was scanned. During the imaging, the absence of lens shadow on B scan was noted to confirm the scan was indeed an avoid-the-lens scan. The gain was again reduced to 0 dB. Static scans of the USS both through the lens and avoiding the lens were recorded. (see Figure 1).

Two authors, RR and KKN, analyzed the fundus pictures and USS images independently. All the scans were analyzed for quality and optic nerve appearance. The scans with well centered optic nerves were selected for analysis. The following features were noted for each disc on fundus photography: the contour of elevation (smoothly elevated or with a ‘lumpy bumpy’ appearance), the number and characteristic of the vessel branching (increased branching, trifurcation of the arteries, cilioretinal arteries, optociliary shunts, and the number of capillaries on the disc surface.

A Fisher’s Exact Test (two-tailed) was used to compare the rate of ODD detection using the through the lens and avoid-the-lens technique. Specificity and sensitivity were also calculated for each technique.

## 3. Results

Thirty-one patients (62 eyes) who presented with bilateral disc edema and met the inclusion criteria were included in the study. Some patients had unilateral drusen and so ‘n’ will refer to total eyes scanned instead of number of patients. The age and refraction of all groups are noted in Table 1. The disc features of all the groups are in Table 2.

Of the 62 eyes: 27 eyes or 15 patients were positive for ODD on USS. Out of the 27 eyes, ODD were detected while scanning through the lens in only in 5 eyes (18%) (mean age 11.5 years., SD = 3.58, Range: 9–16 years.). In the remaining 22 eyes (82%) the ODD were only detected when the scan was done avoiding the lens (*p* = 0.0004) (mean age 14.22 years, SD = 1.7, Range: 12–17 years) (Figure 1). The difference in the age between these two subsets was not statistically significant (*p* = 0.08). 

The patients were divided into three groups: the drusen group, the disc edema group, and the pseudo papilledema group. The drusen group had all of the patients where the drusen were detected through the lens or avoiding the lens. The disc edema group included patients with optic nerve swelling secondary to an identifiable cause. The pseudopapilledema group had no drusen on USS and patients had no identifiable cause for the swelling (there was no evidence of raised ICP on lumbar puncture); they were all hypermetropic on refraction.

Twenty of 27 eyes with ODD on US were noted to have blurred disc margins and elevated discs. While 25 out of 27 (92%) eyes had anomalous vessels over the disc (Figure 2a), only 5 eyes had a disc that appeared lumpy bumpy and three eyes had surface ODD (Figure 2c).

Thirteen eyes were hyperopic, and 14 eyes were emmetropic. The mean refraction was +1.15 D (SD = 1.42, Range: 0–+4.5 D).

Out of the 16 patients who did not have ODD, 10 (20 eyes) had optic disc edema due to secondary cause and were included in disc edema group. Fourteen eyes had disc edema secondary to increased intracranial pressure, four eyes had disc edema secondary to sleep apnea, and two eyes had disc swelling secondary to Kawasaki Disease. In this group of 10 patients, 6 presented with headache and 2 were asymptomatic but were found to have the disc swelling as an incidental finding. In one patient disc swelling was noted when he had an ophthalmology evaluation due to his Kawasaki disease. Although 15 eyes had anomalous vessels over the disc surface (Figure 3), none of the optic nerves had a lumpy or bumpy appearance or calcification on the disc surface.

The pseudopapilledema group comprised of 6 patients (12 eyes). All patients in this group were hyperopic with a mean refractive error of +2.93 D (SD = 2.13, Range: +0.5–5.75 D). Two of these 6 patients presented with headache and 7 eyes had anomalous vessels over the disc. 

If we assume that in fact all 12 of these eyes did have ODD that were not detected by either technique, then the sensitivity and specificity of scanning through the lens to detect ODD is 12% and 100%, respectively, while the sensitivity and specificity of scanning avoiding the lens to detect ODD is 69% and 100%, respectively. If however, we assume that only 6 of the eyes with pseudopapilledema did in fact have ODD undetected by either USS technique then the sensitivity and specificity of scanning through the lens to detect ODD is 15 % and 100%, respectively, while the sensitivity and specificity of scanning avoiding the lens to detect ODD is 82% and 100%, respectively. Finally, if we assume that none of the eyes with pseudopapilledema had ODD then the sensitivity and specificity of scanning through the lens to detect ODD is 18% and 100%, respectively, while the sensitivity and specificity of scanning avoiding the lens to detect ODD is 100% and 100%, respectively.

## 4. Discussion

Diagnosing ODD can be difficult, especially in children. The presenting complaint of headache may be of limited utility in differentiating pathologic causes of disc edema from ODD. Neudorf et al. reported headache to be only 64% sensitive and 54% specific in patients with swollen discs [7]. We had patients in all groups studied presenting with headache. 

The appearance of the optic disc can be used to differentiate ODD from other causes of disc edema. Discs with ODD have been described as crowded, ‘lumpy bumpy’, with calcification [1,6]. In addition, there is an increased incidence of abnormal disc vessels, including trifurcations, anomalous vessels, cilioretinal arteries, and optociliary shunts in patients with ODD [1,8,9,10]. However, in children with ODD, the incidence of irregular disc margins is reported as low as 0.4%. In addition, superficial ODD are reported to be present in only 16% of children [7] and in our series this figure was 11% (3 of 27 eyes). Although abnormal disc vessels are described as one of the features of ODD, abnormal vasculature is also reported in optic discs with papilledema. In our study, while 92% of patients with ODD on USS had abnormal disc vessels, so did 75% of patients with secondary causes of disc edema, 59% of those with pseudopapilledema. Thus, the appearance of the disc vessels may not be an absolute indicator of etiology in patients with optic disc swelling either (Figure 2).

Given that the presence of headaches and optic disc appearance may not be reliable indicators for the etiology of optic disc swelling in certain cases, it is paramount to pursue appropriate investigations including USS.

Ultrasound has been described as a sensitive test in differentiating papilledema from pseudopapilledema [7]. Optic disc drusen on USS appear as hyperechoic lesions due to their calcium content [2], but it is important that the gain be reduced to 0 dB to avoid false positives. The ultrasound waves reflected from the optic nerve head are received by the transducer and the image is processed [6]. The ultrasound image resolution depends on the quality and quantity of waves received by the transducer, which in turn is affected by the various structures the ultrasound waves must pass through while traveling to and from the eye including the crystalline lens [5]. Studies have shown that the human crystalline lens attenuates ultrasound waves and consequently decreases the resolution of the images [11] and also diverges the rays causing distortion of images [11]. It is thus desirable to perform USS while avoiding the lens for superior resolution [5]. The avoid-the-lens scans for disc imaging can be performed by asking the patient to look up and placing the ultrasound probe over the lower lid with the marker on the probe in the horizontal plane. This location and orientation of probe is simple to achieve, and thus the lens can be avoided reliably with practice. While established scanning protocols should have some scans that avoid the lens, these protocols require scans that are longitudinal, axial, and transverse, and this established protocol, in the senior authors’ experience, is unnecessarily difficult to complete in children. 

There are reports of difficulty in detecting ODD on USS, more so if they are buried, as these ODD are not calcified enough to be easily detect using established ocular ultrasound protocols [12,13]. However, none of these studies reported whether scanning was performed avoiding the lens. It is imperative to avoid the lens when performing USS, especially in children as the ODD are often buried (89% of eyes in our study) and not as calcified as those found in adults. In our study, the simple maneuver of avoiding the lens increased the detection rate of ODD by 5.5 times. In 82% of cases of ODD, conclusive USS result would not have been achieved if the lens had not been avoided. This may or may not be significant for adults, not only because the ODD are not buried and are easily visible, but also because with increased calcification any lens attenuation of ultrasound waves is overcome by the density of waves reflected back through the lens to the transducer. It is still prudent to avoid the lens when scanning for ODD to increase the visibility of ODD even in adults because this allows the examiner to get used to the technique in more cooperative adult patients.

Optic disc drusen are acellular deposits of calcium in mitochondria [14]. The etiology still remains unclear, although it has been proposed that a smaller scleral canal predisposes optic nerve axons to altered metabolism and degeneration leading to these calcium deposits [15,16]. However, some studies measuring scleral canal size on OCT have not corroborated this finding [17]. Increased prevalence of ODD among the family members of patients with ODD has also led to the belief that there is a genetic component to ODD formation [18].

Spencer et al. reported cases where patients had only anomalous discs with elevation on presentation and developed ODD on follow-up exam [19]. However, they did not examine the eyes avoiding the lens; it may be that calcified ODD were already there but not detected. 

The patients with pseudopapilledema were four years younger than patients with ODD and so it is possible that these patients with pseudopapilledema may have uncalcified drusen or predisposition to ODD development. These patients have been closely followed to observe if they develop ODD on their follow-up exams, and as yet they have not. In this group, while OCT was attempted, the images were not of good enough quality to definitively make a diagnosis of ODD. When we looked at the sensitivity and specificity of each technique to detect ODD taking into consideration that the 12 eyes , 6 eyes, or 0 eyes might have ODD that were not detectable, the sensitivities were much lower for the technique of scanning through the lens compared to the technique scanning avoiding the lens for each possible scenario. 

In our study, the mean age of patients where ODD were detected through the lens was not significantly different from the rest of the patients where ODD were detected only when the lens was avoided.

Various other tests useful in diagnosing the ODD have been used including OCT [20], fundus autofluorescence (AF), and fundus fluorescein angiography (FFA) [21]. While use of OCT for the diagnosis of ODD and differentiation from papilledema is popular [22], the reliability of OCT in such a differentiation has been questioned [4,23]. Also, in the authors’ experience, younger children are unable to reliably cooperate with the process of OCT. Autofluorescence is a quick and non-invasive diagnostic modality for ODD. However, B scan was shown to be superior to AF [13]. FFA is rarely done for the diagnosis of ODD, especially in children, but if it is preformed then the leakage of dye from the disc seen in disc edema is not seen with ODD [1].

One limitation of the study is that our sample size is small. We only included patients with ODD and the incidence of ODD is low, about 0.4%. Future studies on a larger patient population are warranted to determine the age below which the avoid-the -lens scan is mandatory for the diagnosis of ODD.

## 5. Conclusions

In summary, it is critical to rule out ODD as a cause of disc swelling. Headache and abnormal vasculatures are not unique to ODD and are not indicators of etiology. The ultrasound is a noninvasive bedside test that can rule out ODD as a cause of disc swelling [2]. The technique of avoiding the lens, especially in children, appears to increase the rate of ODD detection. (*p* = 0.0004)

Literature search: The database used was PubMed. The search key words used were: optic nerve drusen, Ultrasound, lens, attenuation, children/pediatrics. The database was searched for publications between 1950 to present.

## Figures and Tables

**Figure 1 jcm-08-01449-f001:**
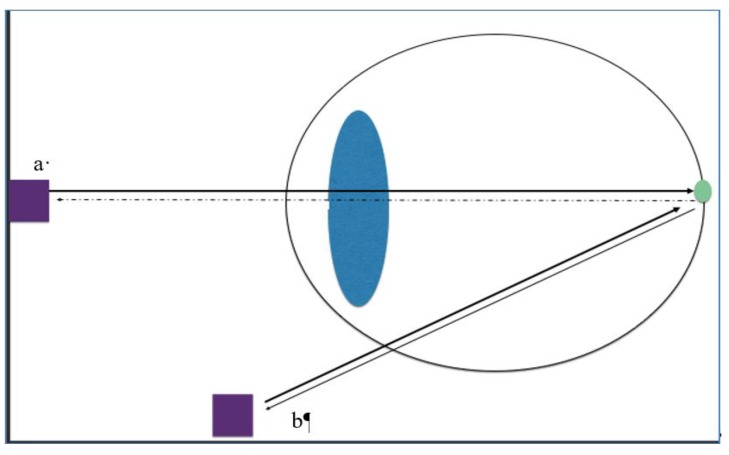
Schematic showing ultrasound scan (USS) probe in position ‘a’ whereby the ultrasound waves travel through the lens ;the ultrasound waves then strike the optic disc drusen (green circle) and are reflected back trhough the lens to be detected by the probe in position ‘a’. If the drusen are buried and have early calcification, the number of reflected ultrasound waves is reduced and these are absorbed by the pediatric lens further reducing the number of ultrasound waves reaching the probe in position ‘a’ (dashed arrow pointing to probe). However, if the probe is positioned on the lower lid with the patient looking up, the lens is avoided and so the reflected ultrasound waves from the optic disc drusen are more easily detected by the probe in position ‘b’ (avoiding the lens) (solid arrow pointing to probe).

**Figure 2 jcm-08-01449-f002:**
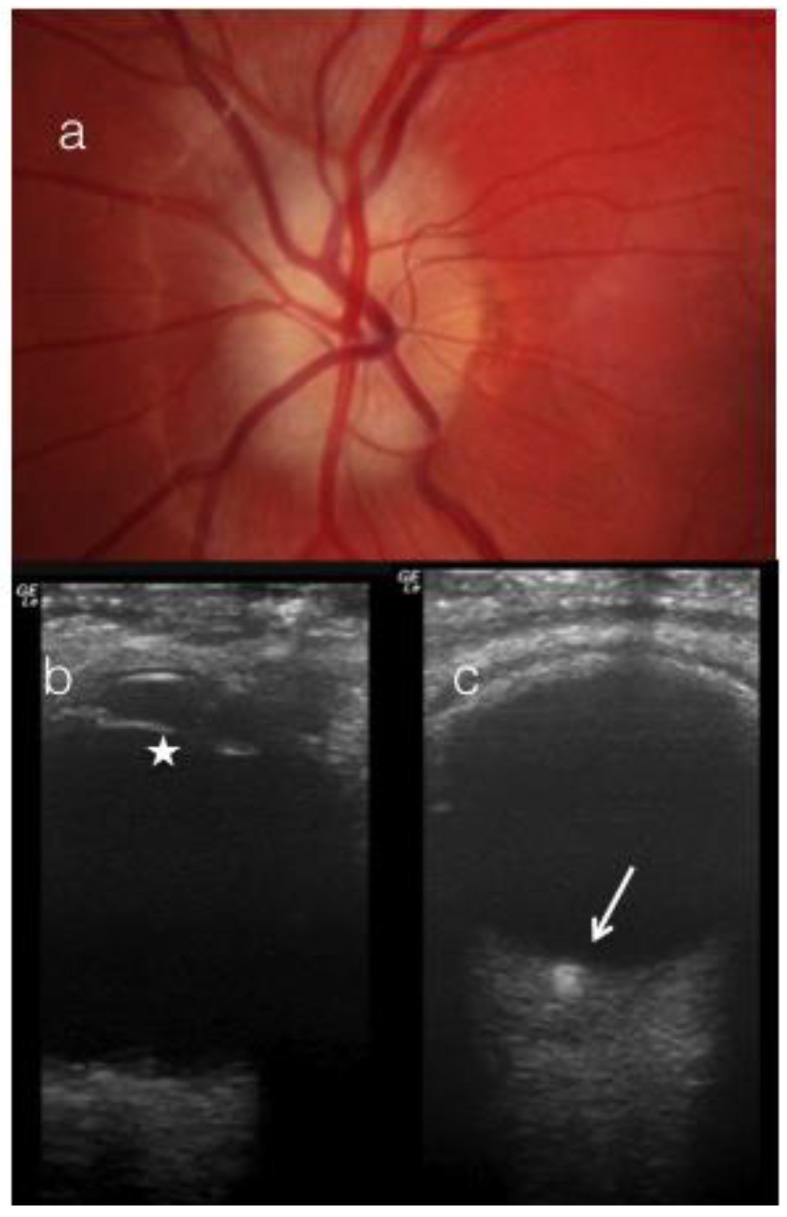
(**a**) Fundus picture of a 15-year. old with optic disc drusen (ODD) (**b**) B scan through the lens (white star) (**c**) B scan avoiding the lens. Note the ODD are more evident in the scan which avoids the lens (white arrow).

**Figure 3 jcm-08-01449-f003:**
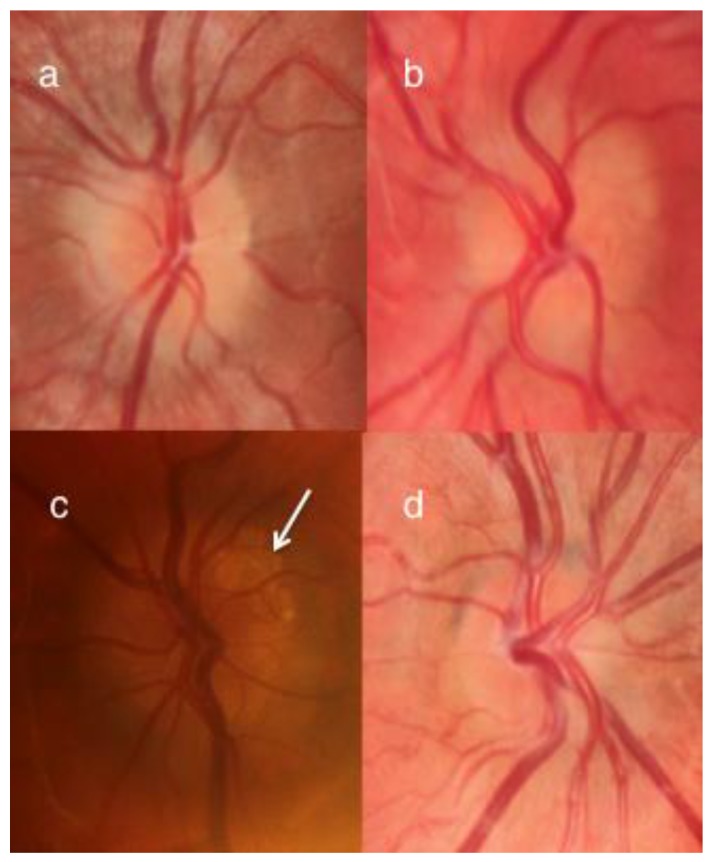
A composite figure showing four optic discs each from different patients (**a**) is a patient with optic disc drusen (ODD) on ultrasound scan (USS); note the anomalous disc vessel branching (**b**) is from a patient with a normal opening pressure on lumbar puncture, negative USS, and known hypermetropia; note the anomalous disc vessel branching (**c**) shows superficial calcification in a disc with ODD (white arrow) (**d**) is a patient with mild disc swelling and raised intracranial pressure on lumbar puncture and anomalous disc vessel branching.

**Table 1 jcm-08-01449-t001:** Details of Patient Cohorts.

Groups	Druse Group	Disc Edema Group	Pseudopapilledema Group
Number (eyes)	27	20	12
Mean age (in years)	13.46	12.3	9.67
Headache	10	12	4
Refraction Mean(D)	+0.97	−0.5	+2.93
Abnormal vessels	25	15	7
Lump bumpy	5	0	0

**Table 2 jcm-08-01449-t002:** Optic Disc Characteristics.

Optic Disc Characteristic	Number (%)
Anomalous vessels	25 (92.5)
Surface drusen	3 (11)
‘lumpy bumpy’ appearance	5 (18)

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
