# Peer review of "Detection of Optic Disc Drusen in Children Using Ultrasound through the Lens and Avoiding the Lens—Point of Care Ultrasound Technique of Evaluation Revisited"

_jcm, 2019, doi:10.3390/jcm8091449_

Round 1
Reviewer 1 Report
The research paper by Rajagopal, R. has presented well the innovative method to detect the rate of ODD with swollen optic discs through the USS crossing the lens. But, there are many points from me to improve the manuscript:
(1) I can not observe any kind of statistical analysis performed throughout the paper.
(2) The number of subjects tested is really low.
(3) Furthemore, the image analysis method should be explicitly explained.
(4) I think it is hard to conclude the facts from such a low number of subjects. This paper can be better explained as a review paper in contrast to a research paper.
Author Response
(1) I can not observe any kind of statistical analysis performed throughout the paper.
RESPONSE :We have applied the Fisher's exact test ( two-tailed) using a contingency table to compare the two techniques ( see methods and results)
(2) The number of subjects tested is really low.
RESPONSE :We acknowledge that the sample size is small. However, the incidence of optic nerve drusen and buried optic nerve drusen is not common. Our study sample size compares with other previously reported optic nerve drusen studies1,2,3. Naoum et al compared multiple modalities of the diagnosis in 19 cases of optic nerve drusen1. Chang et al reported 19 pediatric cases of optic nerve drusen in their study comparing various modalities of diagnosis.3Kulkarni et al reported OCT findings in 15 cases with and without drusen.2
(3) Furthemore, the image analysis method should be explicitly explained.
RESPONSE: we have added to the methods section more detailed information about the technique and further refined the figure legend of the schematic in Figure 1
(4) I think it is hard to conclude the facts from such a low number of subjects. This paper can be better explained as a review paper in contrast to a research paper
RESPONSE: we respectfully disagree. Given the comparable cohorts reported in the literature as cited above , we used the Fisher's exact two tailed test and showed that the difference in detection rate was significant p=0.0004
Reviewer 2 Report
General Editorial Comments:
- Where are the keywords?
- Punctuation needs to be completely and carefully revised throughout the paper (Many of the errors are highlighted in the attached PDF).
- The space before and/or after commas and periods needs to be carefully revised. Also, many times, there are two periods at the end of a sentence.
- If I am not wrong, there are extra spaces between some words in the manuscript. Examples include “calcification and” in Line 36, “the lens” in Line 38, and “80 %” in Line 150-1 (I have highlighted other examples of this possible error in the attached PDF).
- I advise against using numbers in their numerical format at the beginning of sentences unless necessary (for example when reporting ‘x out of y,’ with x and y representing numbers). Also, according to MLA (and other sources), I recommend spelling out numbers under 10 within the text (e.g., Figure 2 caption: A composite figure showing four optic discs …)
- References (within and at the end of the manuscript) are not according to the journal instructions. Please read and follow instructions in https://www.mdpi.com/journal/jcm/instructions. Also, as mentioned above, be careful about the spaces before and after each reference number.
- For figure captions, please make use of the letters a, b, c, etc. in parenthesis, without repeating ‘Figure x#’ in the caption. Please look at Figure 2 in this article for reference: https://www.mdpi.com/2077-0383/8/8/1134/htm.
- In a list of more than two words, place a comma between the last and: x, y, and z.
Specific Editorial Comments:
- In Line 31, 'ultrasound’ needs to be replaced with ‘ultrasound scan’ so that ‘USS’ can correctly represent its abbreviation.
- Please review the grammar of using ‘which’ versus ‘that’ to correct instances like what is highlighted in Line 43 (https://www.writersdigest.com/online-editor/which-vs-that).
- I recommend joining the paragraphs in Line 61 and 62, as the paragraph in Line 61 is disproportionally short.
- Please be consistent in style when reporting statistical values. In the manuscript, sometimes ‘range’ is capitalized (e.g. Line 73) and sometimes not (e.g. Line 71), sometimes a colon is used before numbers (Line 70) and sometimes not (e.g. Line 85). I recommend this format: (Mean: -, SD=-, Range:-). Also kindly be consistent in use of ‘years’ or ‘yrs’ (choose one over the other in all the manuscript).
- Please make sure that ODD is treated as a plural noun all throughout the paper (e.g. in Lines 170-2, ODD is considered a singular noun.)
- Please correct H 2 0 (which contains a zero instead of the letter ‘O’) as follows: H2O.
- Last sentence in Figure 1 caption should be revised as follows, as the use of ‘now’ is not clear: 'Note the ODD are more evident in the scan avoiding the lens'.
- When referring to ‘Group x’ or ‘Group y,’ make sure the first letter is capitalized. This has been nicely done in Lines 109-10, but has been disregarded many other instances (the rule is that when a noun is labeled by a number (like Figure 9), its first letter should be capitalized, as labeling makes it particular noun).
- Please rewrite the sentence in Lines 113-4, as its structure doesn’t sound right.
- Please replace the verb ‘perform’ in Line 141 (‘achieve’ is a good substitute) as location and orientation cannot be performed.
- ‘more so’ seems unnecessary in Line 146.
- Please make the following correction in Line 149: ‘as adults’ Þ ‘as in adults.’
- 'Scleral canal predisposes' WHAT 'to altered metabolism?'
- Please hyphenate ‘follow up’ when used as an adjective as in Line 167.
- Please make the following correction in Line 168: ‘yet have not’ Þ ‘yet they have not.’
- As OCT is defined in Line 31, it does not need to be defined again in Line 174.
- ‘optic nerve ODD [optic nerve optic disc drusen]’ sounds redundant in Line 178, as it contains two instances of ‘optic.’
- Please make the following correction in Line 180: ‘the sensitivity is less than USS’ Þ ‘its sensitivity is less than that of USS.’
- Please make the following correction in Line 183: ‘with’ Þ ‘in.’
- Please either use ‘noninvasive’ (without hyphenation) or ‘non-invasive’ (with hyphenation).
- Please capitalize the first letter of names in Author Contribution: Christin Sylvester.
- Please make the following correction in Line 197: ‘No conflicting relationships exist for any of the authors.’
Please check the rest of the numerous language errors in the highlights in the attached PDF.
Technical Comments:
- Please make a table to clearly show your results, as following all the various cases reported in the Results section is rather difficult to follow in Section 3.
- Please add a separate section for Conclusions and definitely mention that, according to your findings, headache and appearance of the disc vessels are not indicators of etiology of the mentioned conditions. Also, include your suggested protocol in this section in a clear and instructive sentence.
- You mention that USS detected ODD in 44% of the eyes. How do you make sure that the other 56% did not have ODD (possible negative result)? Did you check it with the fundus pictures? Also, can you do a specificity-sensitivity analysis on the results?
- Statistical analysis needs to be presented more clearly.
- Please discuss at least some of the references and other techniques of ODD detection in the Introduction.

Author Response
General Editorial Comments:
- Where are the keywords?
RESPONSE : These have been added;we apologize for having ommitted them
- Punctuation needs to be completely and carefully revised throughout the paper (Many of the errors are highlighted in the attached PDF).
RESPONSE : thank you so much for your help. These have all been rectified as advised
- The space before and/or after commas and periods needs to be carefully revised. Also, many times, there are two periods at the end of a sentence.
RESPONSE : revised as advised
- If I am not wrong, there are extra spaces between some words in the manuscript. Examples include “calcification and” in Line 36, “the lens” in Line 38, and “80 %” in Line 150-1 (I have highlighted other examples of this possible error in the attached PDF).
RESPONSE : revised as advised
- I advise against using numbers in their numerical format at the beginning of sentences unless necessary (for example when reporting ‘x out of y,’ with x and y representing numbers). Also, according to MLA (and other sources), I recommend spelling out numbers under 10 within the text (e.g., Figure 2 caption: A composite figure showing four optic discs …)
RESPONSE : rectified
- References (within and at the end of the manuscript) are not according to the journal instructions. Please read and follow instructions in https://www.mdpi.com/journal/jcm/instructions. Also, as mentioned above, be careful about the spaces before and after each reference number.
RESPONSE : rectified
- For figure captions, please make use of the letters a, b, c, etc. in parenthesis, without repeating ‘Figure x#’ in the caption. Please look at Figure 2 in this article for reference: https://www.mdpi.com/2077-0383/8/8/1134/htm.
RESPONSE : rectified
- In a list of more than two words, place a comma between the last and: x, y, and z.
RESPONSE : rectified
Specific Editorial Comments:
- In Line 31, 'ultrasound’ needs to be replaced with ‘ultrasound scan’ so that ‘USS’ can correctly represent its abbreviation.
RESPONSE : done
- Please review the grammar of using ‘which’ versus ‘that’ to correct instances like what is highlighted in Line 43 (https://www.writersdigest.com/online-editor/which-vs-that).
RESPONSE : changed as suggested
- I recommend joining the paragraphs in Line 61 and 62, as the paragraph in Line 61 is disproportionally short.
RESPONSE : done
- Please be consistent in style when reporting statistical values. In the manuscript, sometimes ‘range’ is capitalized (e.g. Line 73) and sometimes not (e.g. Line 71), sometimes a colon is used before numbers (Line 70) and sometimes not (e.g. Line 85). I recommend this format: (Mean: -, SD=-, Range:-). Also kindly be consistent in use of ‘years’ or ‘yrs’ (choose one over the other in all the manuscript).
RESPONSE : changed to be consistent
- Please make sure that ODD is treated as a plural noun all throughout the paper (e.g. in Lines 170-2, ODD is considered a singular noun.)
RESPONSE : changed as advised
- Please correct H 2 0 (which contains a zero instead of the letter ‘O’) as follows: H2O.
RESPONSE : done
- Last sentence in Figure 1 caption should be revised as follows, as the use of ‘now’ is not clear: 'Note the ODD are more evident in the scan avoiding the lens'.
RESPONSE : done
- When referring to ‘Group x’ or ‘Group y,’ make sure the first letter is capitalized. This has been nicely done in Lines 109-10, but has been disregarded many other instances (the rule is that when a noun is labeled by a number (like Figure 9), its first letter should be capitalized, as labeling makes it particular noun).
RESPONSE: done
- Please rewrite the sentence in Lines 113-4, as its structure doesn’t sound right.
RESPONSE : done
- Please replace the verb ‘perform’ in Line 141 (‘achieve’ is a good substitute) as location and orientation cannot be performed.
RESPONSE : done
- ‘more so’ seems unnecessary in Line 146.
RESPONSE : deleted
- Please make the following correction in Line 149: ‘as adults’ Þ ‘as in adults.’
RESPONSE : changed as requested
- 'Scleral canal predisposes' WHAT 'to altered metabolism?'
RESPONSE : changed to "The etiology still remains unclear, although it has been proposed that a smaller scleral canal predisposes optic nerve axons to altered metabolism and degeneration leading to these calcium deposits15,16"
- Please hyphenate ‘follow up’ when used as an adjective as in Line 167.
RESPONSE: done
- Please make the following correction in Line 168: ‘yet have not’ Þ ‘yet they have not.’
RESPONSE : done
- As OCT is defined in Line 31, it does not need to be defined again in Line 174.
RESPONSE : changed as advised
- ‘optic nerve ODD [optic nerve optic disc drusen]’ sounds redundant in Line 178, as it contains two instances of ‘optic.’
RESPONSE : omitted
- Please make the following correction in Line 180: ‘the sensitivity is less than USS’ Þ ‘itssensitivity is less than that of USS.’
RESPONSE : done
- Please make the following correction in Line 183: ‘with’ Þ ‘in.’
RESPONSE ; done
- Please either use ‘noninvasive’ (without hyphenation) or ‘non-invasive’ (with hyphenation).
RESPONSE : used noninvasive
- Please capitalize the first letter of names in Author Contribution: Christin Sylvester.
RESPONSE : done
- Please make the following correction in Line 197: ‘No conflicting relationships exist for any of the authors.’
RESPONSE: done
Please check the rest of the numerous language errors in the highlights in the attached PDF.
RESPONSE : thank you so much for your help. These have all been incorporated as suggested
Technical Comments:
- Please make a table to clearly show your results, as following all the various cases reported in the Results section is rather difficult to follow in Section 3.
RESPONSE : altered as advised
- Please add a separate section for Conclusions and definitely mention that, according to your findings, headache and appearance of the disc vessels are not indicators of etiology of the mentioned conditions. Also, include your suggested protocol in this section in a clear and instructive sentence.
RESPONSE : altered and added
- You mention that USS detected ODD in 44% of the eyes. How do you make sure that the other 56% did not have ODD (possible negative result)? Did you check it with the fundus pictures? Also, can you do a specificity-sensitivity analysis on the results?
RESPONSE : Ten patients , 20 eyes had no evdience of ODD on USS but had raised ICP or other documented reasons for optic disc swelling. In 6 patients (12 eyes) no ODD were detected and had no signs of raised ICP. Fundus pictures were checked and clarification in methods and results has been added. Moreover we have calculated specificities and senstivities for both techniques , assuming 3 scenarios: that all of these 12 eyes did in fact of ODD, or that only 6 did have ODD or that in fact none had ODD. The sensitivities are much lower in the technique scanning through the lens than scanning avoiding the lens in any of the three assumed scenarios.
- Statistical analysis needs to be presented more clearly.
RESPONSE : statistical analysis added in methodology
- Please discuss at least some of the references and other techniques of ODD detection in the Introduction.
RESPONSE : this has been added as suggested
Round 2
Reviewer 1 Report
The authors made significant changes to the manuscript according to the comments. The paper can be accepted in the current format.
Author Response
Thank you
No comments to respond to
Reviewer 2 Report
Dear Authors,
Thanks for addressing most of the editorial and technical comments from the previous review. As for the technical aspect, I think the paper is now in an acceptable state.
However, the style of the writing (including punctuation, indentation, hyphenation, etc.) and grammar still needs revision. The research conducted is important and has essential results for the medical community, but as long as the presentation is not as strong as the research itself, I am afraid that the paper will not receive the attention it deserves.
Thus, I strongly recommend for the authors to ask for English consultation from an expert and completely review the style, diction, and grammar of the manuscript text, including the caption of the figures (Please do not merely rely on your own knowledge and skill in English writing and ask for help). As you can see in the attached PDF, the text still suffers from many writing errors (and I have not highlighted all of them), which prevents the manuscript from being published as is.
Also, I appreciate the discussion of other methods of ODD detection in the Discussion section, but I think introduction can be still expanded using a more thorough review of these methods. Figure 1 can be improved to become more eye-catching, if possible.
Thanks

Author Response
Thanks for addressing most of the editorial and technical comments from the previous review. As for the technical aspect, I think the paper is now in an acceptable state.
RESPONSE : Thank you
However, the style of the writing (including punctuation, indentation, hyphenation, etc.) and grammar still needs revision. The research conducted is important and has essential results for the medical community, but as long as the presentation is not as strong as the research itself, I am afraid that the paper will not receive the attention it deserves. Thus, I strongly recommend for the authors to ask for English consultation from an expert and completely review the style, diction, and grammar of the manuscript text, including the caption of the figures (Please do not merely rely on your own knowledge and skill in English writing and ask for help). As you can see in the attached PDF, the text still suffers from many writing errors (and I have not highlighted all of them), which prevents the manuscript from being published as is.
RESPONSE: We have engaged a writer to review and change the manuscript
Also, I appreciate the discussion of other methods of ODD detection in the Discussion section, but I think introduction can be still expanded using a more thorough review of these methods. Figure 1 can be improved to become more eye-catching, if possible.
RESPONSE:We have done as asked